# Understanding the Potential Gene Regulatory Network of Starch Biosynthesis in Tartary Buckwheat by RNA-Seq

**DOI:** 10.3390/ijms232415774

**Published:** 2022-12-12

**Authors:** Juan Huang, Bin Tang, Rongrong Ren, Min Wu, Fei Liu, Yong Lv, Taoxiong Shi, Jiao Deng, Qingfu Chen

**Affiliations:** Research Center of Buckwheat Industry Technology, Guizhou Normal University, Guiyang 550001, China

**Keywords:** Tartary buckwheat, RNA-seq, amylose, amylopectin, starch biosynthesis, DEGs, gene regulatory network, transcription factor

## Abstract

Starch is a major component of crop grains, and its content affects food quality and taste. Tartary buckwheat is a traditional pseudo-cereal used in food as well as medicine. Starch content, granule morphology, and physicochemical properties have been extensively studied in Tartary buckwheat. However, the complex regulatory network related to its starch biosynthesis needs to be elucidated. Here, we performed RNA-seq analyses using seven Tartary buckwheat varieties differing in starch content and combined the RNA-seq data with starch content by weighted correlation network analysis (WGCNA). As a result, 10,873 differentially expressed genes (DEGs) were identified and were functionally clustered to six hierarchical clusters. Fifteen starch biosynthesis genes had higher expression level in seeds. Four trait-specific modules and 3131 hub genes were identified by WGCNA, with the lightcyan and brown modules positively correlated with starch-related traits. Furthermore, two potential gene regulatory networks were proposed, including the co-expression of *FtNAC70*, *FtPUL,* and *FtGBSS1-3* in the lightcyan module and *FtbHLH5*, *C3H*, *FtBE2*, *FtISA3*, *FtSS3-5*, and *FtSS1* in the brown. All the above genes were preferentially expressed in seeds, further suggesting their role in seed starch biosynthesis. These results provide crucial guidance for further research on starch biosynthesis and its regulatory network in Tartary buckwheat.

## 1. Introduction

*Fagopyrum tataricum*, also called Tartary buckwheat, belongs to the *Fagopyrum* genus of *Polygonaceae* Mill and is a type of traditional pseudo-cereal for both medicine and food. Tartary buckwheat is famous for its high nutritional value, i.e., balanced amino acids, vitamins, minerals, and flavonoids (especially rutin and quercetin) [1,2,3]. These compounds have numerous effects and can prevent several chronic human diseases, such as hypertension, obesity, cardiovascular diseases, and gallstone formation [3,4].

Starch, comprised of amylose and amylopectin, is the major component of Tartary buckwheat seeds, accounting for the highest proportion of their dry weight [5]. Starch content, especially amylose content and the ratio of amylose to amylopectin content (amylose/amylopectin), affects food quality and taste to a great degree [6,7,8]. The amylose content of Tartary buckwheat seeds ranges from 10–28%, which depends on the germplasm difference and also differences in measurement method [5,9,10]. For granule morphology, Tartary buckwheat starch has a size ranging from 2–14 µm and three kinds of shapes, namely polygonal, oval, and spherical with smooth surface [11,12]. In terms of physicochemical properties, the amylose of Tartary buckwheat is similar to that of common buckwheat (*F. esculentum*), wheat (*Triticum aestivum*) and barley (*Hordeum vulgare*), with 19.3–20.2% iodine affinity, 1.36–1.48 blue values, and 645–657 nm maximum absorption wavelength [13]. The starch solubility in Tartary buckwheat ranges from 9.9–10.4% at 90 °C, which is much lower than that for maize and potato [12]. Meanwhile, peak viscosity and breakdown of Tartary buckwheat are higher, and its pasting time is shorter than that of common buckwheat. It is suggested that Tartary buckwheat starch could be exploited as a suitable raw material for retrograded starch in food processing [12].

In endosperm, starch biosynthesis is a complex process controlled by the synergistic action of five starch synthetases, namely ADP-glucose pyrophosphorylase (AGPase), granule-bound starch synthase (GBSS), soluble starch synthase (SS), starch branching enzyme (BE), and starch debranching enzyme (DBE) [14,15]. AGPase is a hetero-tetramer formed by two large subunits (AGPL) and two small subunits (AGPS) and is responsible for the synthesis of ADP glucose (the major substrate for starch synthesis), which is the first committed and rate-limiting step in the pathway of starch synthesis [16]. GBSS is required for amylose biosynthesis in the form of oligomers after being phosphorylated on the surface of starch granules. On the other hand, the coordination of SS, BE, and DBE is required for amylopectin biosynthesis [14]. In rice and maize, there are five types of SS existing in the endosperm, among which SSI, SSII, and SSIII are responsible for a-glucan chain elongation during amylopectin synthesis, SSIV is responsible for starch granule initiation with the interaction of other proteins, and SSV is involved in manipulating starch granule initiation [17,18]. BE is responsible for producing branches for amylopectin that are connected by a-1,6-glycoside bonds [15]. DBE, comprised of isoamylase (ISA) and pullulanase (PUL), is responsible for correcting incorrect branches in starch synthesis to ensure the orderly synthesis of amylopectin [15].

Genes encoding the starch biosynthesis enzymes have been massively characterized and functionally analyzed in cereals, such as rice (*Oryza sativa*), maize (*Zea mays*), wheat, and barley [14,15]. To date, 3, 4, 2, 9, 4, 3, and 1 gene(s) in rice, and 4, 4, 3,10, 4, 3, and 1 gene(s) in maize that encode AGPS, AGPL, GBSS, SS, BE, ISA, and PUL, respectively, have been identified and their functions studied in depth [15,19,20]. In recent years, focus has gradually been transferred to the regulation mechanism and the regulatory network of starch biosynthesis that is controlled by the regulators, especially the transcription factors (TFs) [21]. Numerous TFs, including bZIP, AP2/ERF, bHLH, NAC, MYB, GRAS, WRKY, MADS, and DOF, are involved in the regulatory network of starch biosynthesis [21]. On one hand, some of the TFs mediate the expression of starch biosynthesis genes through directly binding to the cis-acting elements in their promoters. These TF families include NAC, AP2/ERF, bZIP, MYB, DOF, GRAS, and WRKY [22,23,24,25,26]. A popular example in maize is *ZmNAC128* and *ZmNAC130*, which can bind to the prompters of *Brittle 2* (*BT2*), *Zpu1* (encoding DBE), *GBSSI*, *Sh2* (encoding AGPL), *SSV*, *ISA2*, and *SSIIa*, and thus positively affect starch synthesis in the endosperm of maize [22]. On the other hand, some TFs cannot directly bind to the promoters of the starch biosynthesis genes. They bind to the promoters of the TFs, which act in a direct manner, thus forming a regulatory complex to mediate the expression of starch biosynthesis genes in an indirect manner. These TF families include bHLH, NAC, AP2/ERF, bZIP, MADS, and DOF [21,27,28,29]. For example, a novel FLOURY ENDOSPERM2 (FLO2) could interact with the bHLH proteins in a transcriptional complex that can regulate storage starch by influencing the development of endosperm in rice [27].

In Tartary buckwheat, starch-related research has focused on the starch content, granule morphology, and physicochemical properties [9,11,12,13]. The complex regulatory network of starch biosynthesis in Tartary buckwheat seeds needs to be elucidated. Up to now, only one starch-related gene involved in amylose biosynthesis, namely *FtGBSSI* (Genbank accession: KC990539.1), has been isolated [30]. Meanwhile, a transcriptome analysis revealed that 23 differentially expressed genes (DEGs) correspond to starch biosynthesis [31]. These works are not sufficient for us to understand the gene regulatory network of starch biosynthesis. Here, we have selected seven varieties that differ in starch content to perform RNA-seq analyses and identify the starch biosynthesis genes at a global level. Meanwhile, we combined the RNA-seq data with starch content using weighted gene correlation network analysis (WGCNA), and a series of hub genes and the regulatory network including both starch biosynthesis genes and the candidate TFs were proposed for Tartary buckwheat. This work will not only enrich the gene resources related to starch, but also provide a molecular insight into the starch biosynthesis regulatory mechanism in Tartary buckwheat.

## 2. Results

### 2.1. Starch Content among Tartary Buckwheat Seeds

Amylose and amylopectin contents were measured in the matured seeds of 20 Tartary buckwheat varieties (lines), which presented obvious characteristics in their seed size, morphology, shell color, and shell thickness. The results showed that the amylose content ranged from 17.61% to 21.00%, with an average of 19.24%. The amylopectin content ranged from 19.67% to 28.49%, with an average of 23.63%. The ratio of amylopectin/amylose ranged from 1.05 to 1.43, with an average of 1.23 (Appendix A). We subsequently picked seven varieties that differed in starch content and measured their amylose and amylopectin contents in late-filling-stage seeds, namely PK1, BK2, QK2, XMQ, DMQ, JQ2, and M11 (see full name in Materials and Methods). The amylose content ranged from 3.34% to 11.84% (Figure 1a). The amylopectin content ranged from 48.39% to 66.98% (Figure 1b). The ratio of amylopectin/amylose ranged from 5.66 to 14.51 (Figure 1c). Interestingly, the amylose content, amylopectin content, and amylopectin/amylose produced a gradient. Among these, PK1 had the lowest amylose content, the lowest amylopectin content, and the highest amylopectin/amylose. Therefore, PK1 was used as the control variety in the following analysis.

### 2.2. Global Analysis of RNA-Seq Data in Tartary Buckwheat Seeds

To discover the global changes in gene expression level, high-throughput RNA-seq was carried out for late-filling-stage seeds of seven selected varieties (PK1, BK2, QK2, XMQ, DMQ, JQ2, and M11), with three biological replicates for each variety. In each sample, 40,261,724–102,065,872 clean reads and 6.04–15.31 G clean bases were obtained. Q30 ranged from 88.86% to 89.77% and GC content ranged from 47.64% to 50.03% (Appendix A). The clean reads were then mapped to the genome of Tartary buckwheat. As a result, 35,065,134–88,503,506 (85.52%–87.99%) clean reads were uniquely mapped, among which 17,430,434–43,984,338 (42.48%–43.81%) were uniquely mapped to the “+” strand, whereas 17,634,700–44,519,168 (43.05%–44.18%) were uniquely mapped to the “−” strand of the coding sequences (Appendix A). After mapping, 24,804–26,719 genes were detected in each library, coming to a total of 29,978 genes (Appendix A). According to the Pearson correlation analysis, the repeatability of the biological replicates was acceptable, with correlations ranging from 0.97 to 1 (Appendix A).

Using PK1 as a control, 10,873 DEGs were identified, among which 9573, 3490, 1655, 977, 909, and 906 were identified in the comparisons of M11_vs._PK1, JQ2_vs._PK1, DMQ_vs._PK1, BK2_vs._PK1, QK2_vs._PK1, and XMQ_vs._PK1, respectively (Figure 2a). Of these, 307 DEGs were common in all six comparisons. In addition, 119 DGEs were common in five comparisons except for BK2_vs._PK1; 39 DGEs were common in five comparisons except for QK2_vs._PK1; 34 DGEs were common in five comparisons except for XMQ_vs._PK1; 17 DGEs were common in five comparisons except for M11_vs._PK1; and 17 DGEs were common in five comparisons except for JQ2_vs._PK1 (Figure 2a). Meanwhile, the hierarchical cluster analysis classified all DEGs to six clusters, named as C1 to C6 (Figure 2b,c). Among these, C1 included 996 genes that were highly expressed in PK1, BK2, and QK2, but minimally expressed in JQ2; C2 included 4649 genes that were most highly expressed in PK1, BK2, and QK2 but least expressed in M11; C3 included 210 genes that were least expressed in PK1; C4 included 3476 genes that were highly expressed in M11 but minimally expressed in the other five varieties; C5 included 212 genes that were least expressed in PK1 and M11; C6 included 1327 genes that were highly expressed in JQ2 but minimally expressed in PK1.

The metabolic pathways that the DEGs were related to were analyzed based on KEGG annotation. Among the top 20 enriched pathways in the six comparisons, four metabolic pathways, namely “starch and sucrose metabolism”, “biosynthesis of secondary metabolites”, “circadian rhythm—plant”, and “metabolic pathways” were enriched in all six comparisons (Appendix A). Interestingly, the value of the factor “starch and sucrose metabolism” increased in the enrichment of DEGs identified by comparing the varieties containing higher starch content to PK1, such as M11_vs._PK1 and JQ2_vs._PK1. This suggested that the “starch and sucrose metabolism” pathway was positively related to starch content.

Meanwhile, the identified DEGs in each comparison were annotated to a KOG category. As a result, “carbohydrate transport and metabolism”, “secondary metabolite biosynthesis”, “posttranslational modification, protein turnover, chaperones”, “transport and catabolism”, “signal transduction mechanisms”, and “general function prediction only” were listed in the top annotated categories in all comparisons (Appendix A). The category “carbohydrate transport and metabolism” included the process of starch biosynthesis and metabolism, which was consistent with “starch and sucrose metabolism” enriched in the KEGG pathway.

We also performed GO analysis for the DEGs and the top 50 enriched items related to biological processes, cellular components, and molecular functions were identified (Appendix A). All common items in six comparisons belonged to biological processes. Among these, “secondary metabolic process” was enriched in all six comparisons; “response to toxic substances”, “response to hydrogen peroxide”, “killing of cells of other organisms”, “disruption of cells of other organisms”, and “cell killing” were enriched in five comparisons except for M11_vs._PK1.

### 2.3. Expression of Genes Related to the Starch Biosynthesis Pathway

In plants, AGPases are hetero-tetramers formed by two AGPLs and two AGPSs that catalyze the first committed and rate-limiting step in the starch synthesis pathway [16]. Then GBSS is required for amylose biosynthesis, meanwhile, the coordination of SS, BE, and DBE is required for amylopectin biosynthesis [14]. We identified all genes related to starch biosynthesis pathway at the genome-wide scale previously [32], resulting in 6 ADPase encoding genes (*FtAGPL1~FtAGPL4* and *FtAGPS1~FtAGPS2*), 5 GBSS encoding genes (*FtGBSS1~FtGBSS5*), 12 SS encoding genes (*FtSS1*, *FtSS2-1*~*FtSS2-3*, *FtSS3-1*~*FtSS3-5*, and *FtSS4-1*~*FtSS4-3*), 4 BE encoding genes (*FtBE1*~*FtBE4*), and 4 DBE encoding genes (*FtISA1*~*FtISA3* and *FtPUL*) (Appendix A). Their expression patterns were analyzed based on the transcriptome data. As a result, all 31 genes above were characterized and expressed in seven Tartary buckwheat seeds (Figure 3). Among these, 15 genes, including 3 ADPase encoding genes (*FtAGPL2*, *FtAGPL4*, and *FtAGPS1*), 3 GBSS encoding genes (*FtGBSS1-2*, *FtGBSS1-3*, and *FtGBSS1-5*), 5 SS encoding genes (*FtSS1*, *FtSS2-1*, *FtSS3-2*, *FtSS3-5*, and *FtSS4-3*), 2 BE encoding genes (*FtBE1* and *FtBE2*), and 2 DBE encoding genes (*FtISA3* and *FtPUL*) had higher expression levels, suggesting they had potential functions in the starch biosynthesis of Tartary buckwheat seeds. In addition, 20 out of the 31 genes in the starch biosynthesis pathway were identified as DEGs, including 12 in the hierarchical cluster of C2, 4 in C4, and 4 in C6 (Appendix A, Figure 3). It is worth mentioning that all four genes in C6 (*FtAGPL2*, *FtGBSS1-3*, *FtGBSS1-5*, and *FtPUL*) were more highly expressed in the varieties with higher starch content (Figure 2c).

### 2.4. Trait-Specific Modules and Hub Genes Identified by WGCNA Analysis

WGCNA was performed using 22,191 identified genes whose average count number was equal to or larger than 10 in at least one variety. As a result, nine modules were identified based on the association of gene expression levels with starch-related traits (Figure 4a). Among these, four modules (greenyellow, pink, lightcyan, and brown) were identified as trait-specific modules at the *p* value < 0.05 level. The greenyellow and pink modules were negatively correlated to the amylose content, amylopectin content, and amylose/amylopectin, but positively correlated to amylopectin/amylose. On the other hand, the lightcyan module was positively correlated to amylose content, amylopectin content, and amylose/amylopectin, but negatively correlated to amylopectin/amylose. The brown module was positively correlated to amylose content, amylopectin content, and amylose/amylopectin. Gene expression patterns of four trait-specific modules are shown in Figure 4b. The expression level of genes in the greenyellow module decreased with an increase in the amylose and amylopectin content in seven varieties. The expression level of genes in the pink module also decreased with an increase in the amylose and amylopectin content in six varieties, except for M11. The expression level of the genes in the lightcyan module were lower in PK1 and BK2 but higher in JQ2. The expression level of the genes in the brown module were extremely high in M11 but low in PK1 and BK2.

The hub genes were then identified using the threshold of the top 30% in intramodule connectivity, absolute KME (module eigengene-based connectivity) value ≥0.8, and absolute GS (gene trait significance) value ≥0.2, resulting in a total of 3131 hub genes in four trait-specific modules, including 2409, 217, 415, and 90 hub genes in greenyellow, pink, brown, and lightcyan, respectively (Appendix A).

KEGG pathway analysis was performed for the hub genes in four trait-specific modules and the significantly enriched pathways were screened at a corrected *p*-value < 0.05 level, resulting in 43, 15, 2, and 5 pathways, which were enriched in the greenyellow, pink, lightcyan, and brown modules, respectively (Appendix A). In the greenyellow module, most of the significantly enriched pathways were related to metabolism, in particular to fatty acid biosynthesis (00061), amino acid biosynthesis (valine, leucine, and isoleucine biosynthesis, 00290; lysine biosynthesis, 00300; arginine biosynthesis, 00220; phenylalanine, tyrosine, and tryptophan biosynthesis, 00400), N-glycan biosynthesis (00510 and 00513), energy metabolism, transport, and recycling (A09100; B09102; pyruvate metabolism, 00620; glycolysis/gluconeogenesis, 00010; citrate cycle (TCA cycle), 00020; carbohydrate metabolism, B09101; amino acid metabolism, B09105, 00260, and 00340). Others were related to genetic information processing (A09120, B09122, 03010, 03050, B09123, and 00970), brite hierarchies (03011, 04147, B09182, 03036, 03012, 01007, A09180, 02044, 02048, 03019, and 03051), and cellular processes (phagosome, 04145) (Appendix A). In the pink module, the enriched pathways were divided into two classes, one of which was related to genetic information processing, including genetic information processing (A09120), proteasome (3050), ribosome biogenesis in eukaryotes (3008), translation (B09122), spliceosome (3040), transcription (B09121), folding, sorting, and degradation (B09123), mRNA surveillance pathway (3015), and nucleocytoplasmic transport (3013); the other was related to brite hierarchies, including protein families: genetic information (B09182), ribosome biogenesis (3009), brite hierarchies (A09180), proteasome (3051), spliceosome (3041), and messenger RNA biogenesis (3019) (Appendix A).

In the lightcyan module, no pathway was enriched at a corrected *p*-value < 0.05 level. However, two metabolism-related pathways, namely starch and sucrose metabolism (00500) and carbohydrate metabolism (B09101) were listed on the top two enriched pathways (corrected *p*-value < 0.1, Appendix A and Figure 4c). Two genes in the starch biosynthesis pathway, *FtGBSS1-3* and *FtPUL*, were in this module (Appendix A). In the brown module, the significantly enriched pathways were carbohydrate metabolism (B09101), starch and sucrose metabolism (00500), metabolism (A09100), and galactose metabolism (00052) (Appendix A and Figure 4c). Four genes in the starch biosynthesis pathway, *FtBE2*, *FtSS3-5*, *FtISA3*, and *FtSS1*, were in this module (Appendix A).

### 2.5. Identification of TFs in Four Trait-Specific Modules and Their Co-Expression Network Related to the Starch Biosynthesis Pathway

TFs in the hub genes of four trait-specific modules were screened; the resulting 60, 1, 4, and 36 TFs were enriched in the greenyellow, pink, lightcyan, and brown modules, respectively (Appendix A and Figure 5a). In the greenyellow module, a total of 60 TFs were classified to 40 TF families, including 6, 6, 4, and 4 TFs in the bHLH, MYB, C2C2-GATA, and zf-HD families, respectively. In the pink module, only one bZIP TF (FtPinG0006546100.01) was screened. In the lightcyan module, four TFs were screened, namely two NAC (FtPinG0002339300.01, named as *FtNAC70*, and FtPinG0009656900.01, named as *FtNAC77/15*) [33,34,35], one AP2/ERF-ERF (FtPinG0002070100.01), and one RWP-RK (FtPinG0002032100.01). In the brown module, a total of 36 TFs were classified to 22 TF families, including 5, 3, and 3 TFs in AP2/ERF-ERF, bHLH, and the CCCH-type Zn-finger (C3H) protein family, respectively.

TFs are often in the upstream of a regulatory network, so we identified the TFs in the hub genes of the top 50 in intramodule connectivity, resulting in nine TFs in total, including four TFs in the lightcyan module and five TFs in the brown module (Appendix A). Interestingly, all four TFs in the lightcyan module, namely *FtNAC70* and *FtNAC15*, one AP2/ERF-ERF (FtPinG0002070100.01), and one RWP-RK (FtPinG0002032100.01), were clustered to C6 based on their gene expression patterns; meanwhile, all five TFs in the brown module, namely one E2F-DP (FtPinG0001244300.01), two bHLH (FtPinG0000281400.01, named as *FtbHLH5*, and FtPinG0005387300.01, named as *FtbHLH91*) [36], one WRKY (FtPinG0005111700.01, named as *FtWRKY39*) [37], and one C3H-type Zn-finger protein (FtPinG0003310400.01), were clustered to C4 based on their gene expression patterns.

Nine TFs in the hub genes of the top 50 in intramodule connectivity were taken to perform a co-expression network with the starch biosynthesis genes in the hub genes of the top 30% in intramodule connectivity in the same module. The result showed that one TF, *FtNAC70*, was co-expressed with two starch biosynthesis genes, *FtPUL* and *FtGBSS1-3*, in the lightcyan module (Figure 5b). Meanwhile, two TFs, *C3H* and *FtbHLH5*, were co-expressed with four starch biosynthesis genes, *FtBE2, FtISA3, FtSS3-5,* and *FtSS1*, in the brown module (Figure 5c).

Pearson correlations of the genes in two co-expression networks were calculated. The result showed that three genes in the lightcyan module, *FtNAC70*, *FtPUL,* and *FtGBSS1-3*, were positively correlated to each other at a significance level of 0.01 (Pearson R ≥ 0.79, Figure 5d). Two TFs in the brown module, *FtbHLH5* and *C3H*, were positively correlated to each other (Pearson R = 1) but negatively correlated to four starch biosynthesis genes, *FtBE2, FtISA3, FtSS3-5,* and *FtSS1* (Pearson R ≤ −0.91), at the significance level of 0.01 (Figure 5e). 

### 2.6. Tissue-Specific Expression Patterns of Nine Candidate Genes in the Starch Biosynthesis Pathway

Nine candidate genes identified by the co-expression network in the lightcyan and brown modules (Figure 5b,c) were applied to perform a tissue-specific expression analysis. The result showed that all of the nine genes were preferentially expressed in seeds, rather than the other four tissues (roots, stems, leaves, and flowers) (Figure 6). Among these, *FtNAC70*, *C3H*, and *FtSS3-5* were highly expressed in seeds and minimally expressed in the other four tissues; *FtbHLH5*, *FtGBSS1-3*, and *FtBE2* were highest expressed in roots, followed by stems, and minimally expressed in the other three tissues; *FtSS1*, *FtISA2*, and *FtPUL* were highly expressed in seeds and moderately expressed in the other four tissues.

## 3. Discussion

Seeds are the edible organs of plants, and starch accounts for the highest proportion of the dry weight of seeds in Tartary buckwheat [12]. A food’s quality and taste are greatly influenced by its starch content, especially amylose and the ratio of amylose to amylopectin [6,7,8]. Previous studies have reported that the amylose content of Tartary buckwheat seeds ranged from 10% to 28% [5,9,10]. The amylose content in the seeds of 20 Tartary buckwheat varieties (lines) ranged from 17.61% to 28.49%, which was consistent with previous studies. In addition, our results showed that the ratio of amylose to amylopectin content differed among the varieties (lines), which provides a basis for selecting varieties with different ratios of amylose to amylopectin for the use of Tartary buckwheat in food processing.

RNA-seq is a well-established and widely used method for identifying global gene regulation at a transcriptional level. We carried out RNA-seq analyses for seven varieties that differed in their starch content, resulting in 29,978 mapped genes and 10,873 DEGs, using PK1 as a control. Through metabolic pathway enrichment analysis, the “starch and sucrose metabolism” pathway was found to be significantly enriched and was positively related to starch content. Meanwhile, “carbohydrate transport and metabolism”, which included the process of starch biosynthesis and metabolism, was listed in the top annotated KOG categories. The KOG categories result was consistent with the KEGG enrichment result, suggesting the starch biosynthesis pathway differed in the seven varieties. In recent transcriptome analyses of developing Chinese chestnut (*Castanea mollissima* Blume) seed kernels, genes involved in starch biosynthesis, including AGPase, GBSS, SS, SBE, and ISA encoding genes, were also enriched in the developing seed [38,39], which were consistent with our result for Tartary buckwheat seed.

Based on our RNA-seq data, we identified 31 genes related to the starch biosynthesis pathway, including *FtAGPL1*~*FtAGPL4*, *FtAGPS1*~*FtAGPS2*, *FtGBSS1*~*FtGBSS5*, *FtSS1*, *FtSS2-1*~*FtSS2-3*, *FtSS3-1*~*FtSS3-5*, *FtSS4-1*~*FtSS4-3*, *FtBE1*~*FtBE4*, *FtISA1*~*FtISA3*, and *FtPUL*. Of these, 15 genes, namely *FtAGPL2*, *FtAGPL4*, *FtAGPS1*, *FtGBSS1-2*, *FtGBSS1-3*, *FtGBSS1-5*, *FtSS1*, *FtSS2-1*, *FtSS3-2*, *FtSS3-5*, *FtSS4-3*, *FtBE1*, *FtBE2*, *FtISA3*, and *FtPUL*, had a higher expression level, suggesting they might be responsible for starch biosynthesis in seeds rather than other tissues. To date, only one starch-related gene, namely *FtGBSSI* (Genbank accession: KC990539.1), has been isolated and involved in amylose biosynthesis [30]. This gene was referred to as *FtGBSS1-2* and was highly expressed in the seeds in our study. Another reported starch-related gene in *F. esculentum*, namely *FeGBSS* (Genbank accession: HW041459.1), was highly homologous to *FtGBSS1-1*; however, *FtGBSS1-1* was not preferentially expressed in the seeds in our study, suggesting it might participate in amylose biosynthesis in other tissues than the seeds of Tartary buckwheat. In addition, we previously performed a transcriptome analysis of three seed developmental stages in Tartary buckwheat and identified 23 DEGs corresponding to starch biosynthesis, among which six DEGs were expressed positively with seed development and showed higher expression in the later developmental seeds [31]. At this time, we selected seven varieties that differed in starch content for transcriptome analysis and identified 31 genes corresponding to starch biosynthesis, among which 10 genes showed higher expression in the varieties with high starch content. This work would largely enrich the starch-related gene resources for Tartary buckwheat.

With the combination of the four starch-related traits and the gene expression data achieved through WGCNA analysis, nine modules were identified, of which four modules (greenyellow, pink, lightcyan, and brown) were identified as trait-specific modules. Though the greenyellow and pink modules included more hub genes and more enriched pathways, the starch-related pathway was absent in them but significantly enriched in the lightcyan and brown modules, which were positively correlated to amylose content, amylopectin content, and amylose/amylopectin based on the module-trait relationship. Among the starch biosynthesis genes, *FtGBSS1-3* and *FtPUL* were listed in the lightcyan module and *FtBE2*, *FtSS3-5*, *FtISA3*, and *FtSS1* were listed in the brown module.

According to the results of TF analysis, 4 and 36 TFs were identified in the lightcyan and brown modules, respectively, among which nine TFs were in hub genes of the top 50 in intramodule connectivity. Some of the above TFs, such as NAC, AP2/ERF-ERF, bHLH, and WRKY, were homologous to the functionally characterized regulators related to starch metabolism in cereal crops. For example, NAC family genes are involved in starch biosynthesis in rice and maize. *ZmNAC128* and *ZmNAC130* could bind to the prompters of the starch biosynthesis genes, including *Brittle 2* (*BT2*), *Zpu1* (encoding DBE in maize), *GBSSI*, *Sh2* (encoding AGPL), *SSV*, *ISA2*, and *SSIIa*, and thus positively affect starch synthesis in the endosperm of maize [22]. *TaNAC019* was reported to regulate starch biosynthesis by the regulation of *SSIIa* and *Susy1*, and thus accelerate starch accumulation in wheat [23]. In our study, two NAC TFs, namely *FtNAC70* and *FtNAC15*, were identified as top-50 hub genes. Both of their expression patterns have been reported to be increasing with the seed development of Tartary buckwheat [33,34]. It is worth mentioning that the tissue-specific expression analysis of *FtNAC70* showed that it was a seed-specific NAC TF. Meanwhile, its expression was significantly positively correlated to *FtGBSS1-3* (Pearson R = 0.84) and *FtPUL* (Pearson R = 0.79), suggesting that *FtNAC70* should be a key positive regulator in regulating starch biosynthesis in Tartary buckwheat.

TFs in the bHLH family could regulate starch biosynthesis in an indirect manner. A novel FLOURY ENDOSPERM2 (FLO2) could interact with the bHLH proteins in a transcriptional complex that could regulate starch storage by influencing the development of endosperm in rice [27]. OPAQUE11, an endosperm-specific bHLH TF, is a central hub of the regulatory network for maize endosperm development and nutrient metabolism, a mutant of which showed decreased starch and protein with a small and opaque endosperm [28]. In our study, two bHLH TFs, namely *FtbHLH5* and *FtbHLH91*, were identified as top-50 hub genes. *FtbHLH5* was more highly expressed in seeds and stems, whereas *FtbHLH91* was more highly expressed in leaves rather than other tissues [36]. In addition, both of the two bHLH TFs were increasingly expressed with seed development [36], suggesting that *FtbHLH5* might be involved in regulating starch biosynthesis in seeds and stems, whereas *FtbHLH91* might be involved in regulating starch biosynthesis in the stem. The expression of *FtbHLH5* was significantly negatively correlated to *FtBE2* (Pearson R = −0.96), *FtSS3-5* (Pearson R = −0.93), *FtSS1* (Pearson R = −0.93), and *FtISA3* (Pearson R = −0.92), suggesting that *FtbHLH5* should be a key negative regulator in regulating starch biosynthesis in Tartary buckwheat.

An AP2/ERF-ERF family gene, *ZmEREB156*, was sucrose- and ABA-inducible and positively regulated starch synthesis by directly binding to the promoter of *ZmSSIIIa* in maize [24]. *Sugar Signaling in Barley2* (*SUSIBA2*), a plant-specific WRKY TF, could bind to the promoters of *BEIIb;* the heterologous expression of it in rice produced a rice variety with a high starch content [26]. A mutant of *OS1*, encoding an RWP-RK TF, downregulated certain genes in specific cell types, including a majority of zein- and starch-related genes in central starch endosperm cells [40]. In our study, an AP2/ERF-ERF TF (FtPinG0002070100.01), a WRKY TF (*FtWRKY39*), and an RWP-RK TF (FtPinG0002032100.01), were identified as top-50 hub genes, suggesting they might be involved in starch biosynthesis in Tartary buckwheat. The expression level of *FtWRKY39* increased gradually with the development of seeds [37], suggesting it could regulate starch biosynthesis during seed development.

Though no evidence has been shown at present, an E2F-DP TF (FtPinG0001244300.01) and a C3H-type Zn-finger protein (FtPinG0003310400.01) were also identified as top-50 hub genes, suggesting they might be involved in starch biosynthesis in Tartary buckwheat. Notably, the C3H-type Zn-finger protein was in the co-expression network of *FtbHLH5*, significantly positively correlated to *FtbHLH5* (Pearson R = 1.00) and significantly negatively correlated to *FtBE2*, *FtSS3-5*, *FtSS1*, and *FtISA3*. In addition, *C3H* was preferentially expressed in seeds rather than other tissues. These suggested that *C3H* might interact with *FtbHLH5* and be another key negative regulator in regulating starch biosynthesis in Tartary buckwheat.

## 4. Materials and Methods

### 4.1. Plant Material

A total of 20 Tartary buckwheat varieties/lines were used in this study, namely Biku 2 (BK2), Damiqiao (DMQ), Jinkuqiao (JKQ), Jinqiao 2 (JQ2), Jiujiangkuqiao (JJ), Miku 11 (M11), Miku 127 (M127), Miku 13 (M13), Miku 18 (M18), Miku 5 (M5), Miku 55 (M55), Pinku 1 (PK1), Qianheiqiao 1 (QHQ1), Qianku 2 (QK2), Sanya A5 (SYA5), Sanya B60 (SYB60), Sanya B68 (SYB68), Sanya B69 (SYB69), Sanya C28 (SYC28), and Xiaomiqiao (XMQ). In the early August of 2020, their seeds were sowed in the Anshun experimental field of our laboratory and grew with normal field management. When they were nearly matured, seeds at late filling stage were taken with three replicates, immediately frozen in liquid nitrogen, shelled on ice, and stored in an ultra-low-temperature freezer for starch measurement and RNA-seq. When the plants were matured, the ripened seeds were harvested and dried for starch measurement. Then the ripened seeds of all 20 varieties/lines and the late-filling-stage seeds of seven varieties, namely PK1, BK2, QK2, XMQ, DMQ, JQ2, and M11 were used for starch measurement, with three biological replicates and two (or three) technical replicates. Meanwhile, the late-filling-stage seeds of the seven mentioned varieties were used for RNA-seq analysis, with three biological replicates.

### 4.2. Starch Content Measurement

Amylose and amylopectin content were measured in accordance with a previous report [41], with some modification. Briefly, the amylose and amylopectin standards were brought from Sigma-Aldrich (Merck KGaA, Darmstadt, Germany). To obtain a mixed standard curve, 11 mixed standards with a series of amylose and amylopectin concentrations were made, namely 0, 5, 10, 15, 20, 25, 30, 35, 40, 45, and 50 µg/mL amylose, and 100, 90, 80, 70, 60, 50, 40, 30, 20, 10, and 0 µg/mL amylopectin in every standard, respectively. The pH was adjusted to 3.0 using 2 mol/L HCl. Then, 0.5 mL iodine reagent was added and mixed. After standing at room temperature for 25 min, the mixtures were applied to a spectrophotometer for amylose measurement at 597 nm and 480 nm and amylopectin measurement at 541 nm and 700 nm. The standard curves of amylose and amylopectin were established using OD_(597 nm–480 nm)_ and OD_(541 nm–700 nm)_, respectively. The equation of the standard curve for amylose was
Y = 1.28 × 10^−2^ X − 2.17 × 10^−2^(1)
and the equation of the standard curve for amylopectin was
Y = 2.8 × 10^−3^ X + 0.219 × 10^−1^(2)

For each variety/line of Tartary buckwheat, 0.1000 g ground powder of the seeds was accurately weighed and placed in a 50 mL centrifuge tube. Then, 10 mL absolute ethanol was added and mixed. The amylose and amylopectin were extracted in a water bath at 80 °C for 30 min. After cooling, the samples were centrifuged at 12,000 rpm at 25 °C for 5 min. The supernatant was discarded, and 10 mL KOH (1 mol/L) was added to the retained precipitate. The sample was placed into a water bath at 100 °C for 15 min, followed by being transferred to a freezer at 4 °C. Afterward, a 1 mL sample was transferred into a new 50 mL centrifuge tube and 10 mL ddH_2_O was added. Then, 200 µL solution was placed into a new 2 mL centrifuge tube and 1 mL ddH_2_O was added. The pH was adjusted to 3.0 using 2 mol/L HCl. Then, 20 µL iodine reagent was added and mixed. The volume was titrated to 2 mL with ddH_2_O. After standing at room temperature for 25 min, the mixtures were applied to a spectrophotometer for amylose measurement at 597 nm and 480 nm and amylopectin measurement at 541 nm and 700 nm. Then, the amylose and amylopectin content were calculated using Equations (1) and (2), respectively.

### 4.3. RNA-Seq Analysis

RNA was extracted using the RNAprep Pure Plant Plus Kit produced by TIANGEN (DP441, Beijing, China). Library construction was carried out by Metware Biotechnology Co., Ltd. (Wuhan, China). After that, the library was sequenced on an Illumina NovaSeq 6000 platform (California, USA) and 150 bp paired-end reads were generated. Data filtering and quality control were also completed by this company using fastp, resulting in more than 6 G clean data with Q20 > 95% and Q30 > 85% for each sample [42,43]. Afterward, the clean reads were mapped to the genome data for Tartary buckwheat http://www.mbkbase.org/Pinku1/ (accessed on 8 September 2022) and the chromosomal location of the mapped genes were obtained by HISAT2 with default parameters [44]. The mapped genes were functionally annotated to the databases of NR ftp://ftp.ncbi.nih.gov/blast/db (accessed on 8 September 2022), KEGG https://www.genome.jp/kegg (accessed on 8 September 2022), KOG https://www.ncbi.nlm.nih.gov/COG/ (accessed on 8 September 2022), and GO http://geneontology.org/ (accessed on 8 September 2022) using the BLAST program (e-value was set as 1 × 10^−5^). Gene expression was quantified by FPKM (fragments per kilobase of transcript per million fragments mapped) as the following equation:(3)$$\mathrm{FPKM}=\frac{\mathrm{mapped}\;\mathrm{fragments}\;\mathrm{of}\;\mathrm{transcript}}{\mathrm{total}\;\mathrm{count}\;\mathrm{of}\;\mathrm{mapped}\;\mathrm{fragments}\;\left(\mathrm{millions}\right)\times \;\mathrm{length}\;\mathrm{of}\;\mathrm{fragments}}$$

Pearson correlation was analyzed and visualized using “corrplot” in R language https://cran.r-project.org/web/packages/corrplot/index.html (accessed on 8 September 2022).

DEGs were identified by “DESeq2” in R language with default parameters, using absolute value of Log_2_(FoldChange) ≥1 and padj (adjusted *p*-value) < 0.05 [45]. The upset plot of the DEGs was visualized using “UpSetR” in R language [46]. The hierarchical cluster of the DEGs was performed by “ggplot2”, “pheatmap”, “reshape2”, and “factoextra” in R language https://cran.r-project.org/web/packages/available_packages_by_name.html#available-packages-C (accessed on 8 September 2022). For the hierarchical cluster of all identified DEGs, the scale was set as “row”. For the hierarchical cluster of partial analyzed DEGs, the scale was set as “none”. KEGG pathway and GO item enrichment were performed by TBtools [47]. The top 20 KEGG pathways, top 50 enriched GO items, and KOG categories were visualized by “ggplot2” in R language https://cran.r-project.org/web/packages/ggplot2/index.html (accessed on 8 September 2022).

For tissue-specific analysis, gene expression data of four different tissues (root, stem, leaf, and flower) were obtained from the published genome data for Tartary buckwheat [48], whereas gene expression data for seeds was calculated by mean FPKM value in the seeds of the seven varieties in this study.

### 4.4. WGCNA

WGCNA was performed using “WGCNA” in R language [49]. After filtering the genes whose average count number was less than 10 in every variety, the top 50% mad genes among the remaining genes (11,073 genes) were used for WGCNA analysis. In this study, the parameters were set as softpower = 12, minModuleSize = 30, MEDissThres = 0.2, and deepSplit = 2. The module–trait relationship was made using the combination of gene expression data with four starch-related traits, namely amylose content, amylopectin content, amylose/amylopectin, and amylopectin/amylose. Then, the hub genes were identified by the threshold of top 30% in intramodule connectivity, absolute KME value ≥0.8, and absolute GS value ≥0.2. The TFs in the DEGs were identified using the iTAK plant transcription factor and protein kinase identifier and classifier, http://itak.feilab.net/cgi-bin/itak/online_itak.cgi (accessed on 8 September 2022). The gene co-expression networks of the starch biosynthesis genes and TFs were visualized using Cytoscape [50].

## 5. Conclusions

In this work, we performed RNA-seq analyses using seven Tartary buckwheat varieties that differed in starch content and combined their RNA-seq data with starch content using WGCNA. As a result, 10, 873 DEGs were identified and functionally clustered to six hierarchical clusters based on their expression patterns. Subsequently, starch biosynthesis genes were identified, and 15 of them, including *FtAGPL2*, *FtAGPL4*, *FtAGPS1*, *FtGBSS1-2*, *FtGBSS1-3*, *FtGBSS1-5*, *FtSS1*, *FtSS2-1*, *FtSS3-2*, *FtSS3-5*, *FtSS4-3*, *FtBE1 FtBE2*, *FtISA3* and *FtPUL*, had a higher expression level in seeds. In addition, four trait-specific modules (greenyellow, pink, lightcyan, and brown) and 3131 hub genes in these modules were identified by WGCNA, among which the lightcyan and brown modules were positively correlated to amylose content, amylopectin content, and amylose/amylopectin. Starch and sucrose metabolism and carbohydrate metabolism pathways were enriched in both the lightcyan and brown modules. Based on TF analysis in trait-specific modules, *FtNAC70* was co-expressed with *FtPUL* and *FtGBSS1-3* in the lightcyan module; *FtbHLH5* and *C3H* were co-expressed with *FtBE2*, *FtISA3*, *FtSS3-5*, and *FtSS1* in the brown module. All nine genes in the co-expression networks were preferentially expressed in seeds rather than other tissues, suggesting they might be responsible for starch biosynthesis in the seeds. These results would not only enrich the gene resources related to starch, but also provide a molecular insight into the starch biosynthesis regulatory mechanism in Tartary buckwheat. The results here appear to be an excellent starting point for further research on Tartary buckwheat. This work will be of benefit in the future for a complementary theoretical study using quantum chemical topology to identify the vdW interactions that play a role in the RNA-seq analysis. The estimation of non-covalent interaction energies present in the structures of RNA presented here could help to better guide further research on starch biosynthesis and its regulatory network in Tartary buckwheat.

## Figures and Tables

**Figure 1 ijms-23-15774-f001:**
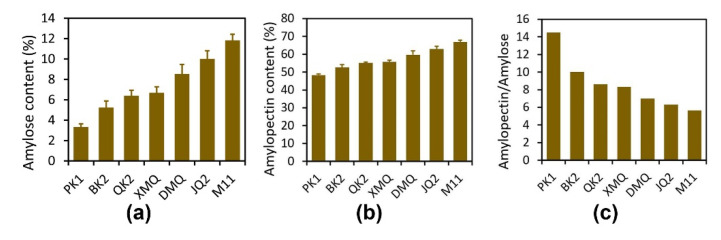
Starch content in late-filling-stage seeds of seven Tartary buckwheat varieties. (**a**) Amylose content; (**b**) amylopectin content; (**c**) amylopectin/amylose. The data represent the mean ± SD that came from three biological replicates and at least two technical replicates for each biological replicate.

**Figure 2 ijms-23-15774-f002:**
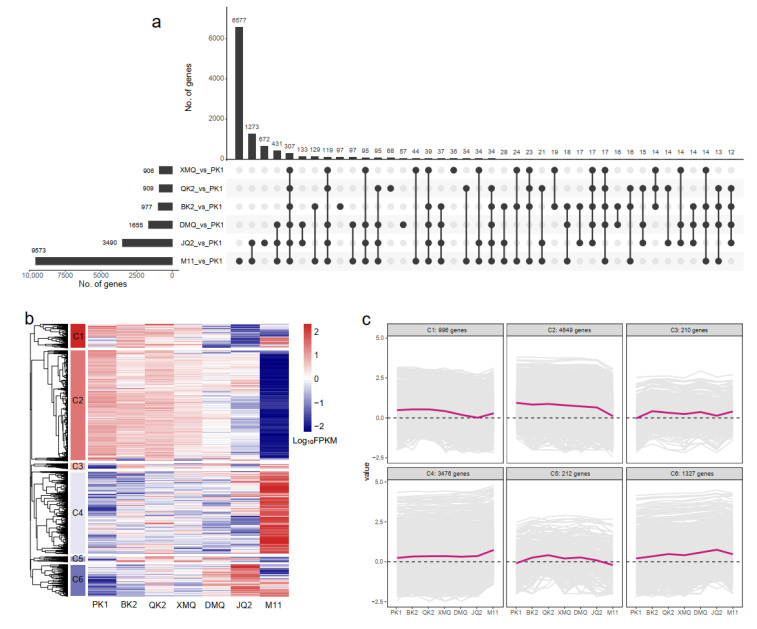
Analyses of differentially expressed genes (DEGs) in seven Tartary buckwheat varieties. (**a**) Upset plot of the DEGs in six comparisons. Black bars on the lower left represent the number of DEGs in comparisons of seven selected varieties. The black dots on the lower right represent the common DEGs existing in six comparisons. Black bars above represent the number of common DEGs in six comparisons; (**b**) Functional category of DEGs by hierarchical cluster; (**c**) Gene expression patterns of the six clusters that correspond to the hierarchical cluster result. Six main clusters were presented as C1–C6. Gene expression values are normalized to log_10_ (FPKM).

**Figure 3 ijms-23-15774-f003:**
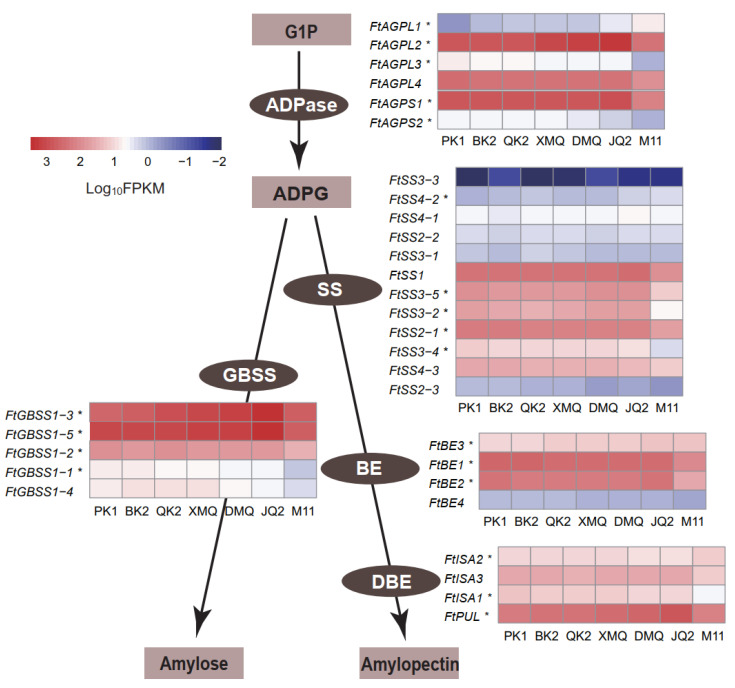
Expression patterns of genes related to the starch biosynthesis pathway. The light-brown rectangles and dark-brown ovals represent the substrates and the enzymes in the starch biosynthesis pathway, respectively. The heatmap shows the expression patterns of genes in the starch biosynthesis pathway. Genes marked with “*” were identified as DEGs. AGPase, ADP-glucose pyrophosphorylase; AGPL, large subunit of AGPase; AGPS, small subunit of AGPase; GBSS, granule-bound starch synthase; SS, soluble starch synthase; BE, starch branching enzyme; DBE, starch debranching enzyme; ISA, isoamylase; PUL, pullulanase.

**Figure 4 ijms-23-15774-f004:**
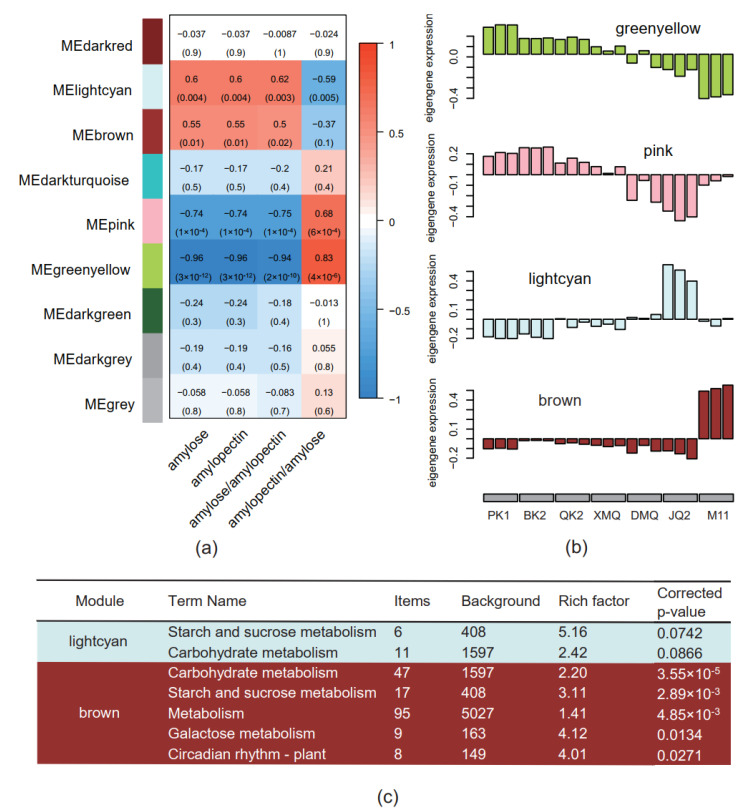
Weighted gene correlation network analysis (WGCNA) for DEGs and the enriched pathways of trait-specific modules in Tartary buckwheat seeds. (**a**) Module–trait relationships by WGCNA analysis. The correlations and the corresponding *p*-values (in parentheses) are indicated in the heatmap. The panel on the left side shows nine identified modules. amylose/amylopectin is shorten for the ratio of amylose to amylopectin; amylopectin/amylose is short for the ratio of amylopectin to amylose; (**b**) Expression patterns of the trait-specific modules (*p*-value < 0.05) in correspondence to the module–trait relationship heatmap; (**c**) KEGG enrichment analysis of the lightcyan and brown modules. Rich factor is the ratio of the number of DEGs to that of all genes annotated to a pathway term. A higher rich factor indicates greater intensity.

**Figure 5 ijms-23-15774-f005:**
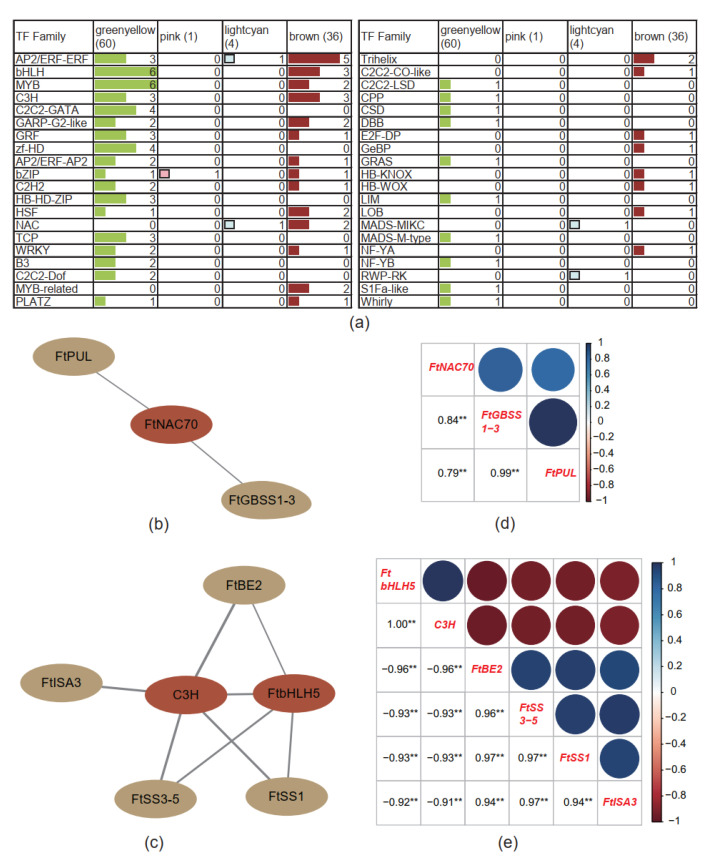
TFs identified in four trait-specific modules and their co-expression networks related to the starch biosynthesis pathway. (**a**) Distribution of transcription factor families in four trait-specific modules. C3H, CCCH-type Zn-finger protein; (**b**,**c**) Co-expression networks of TFs in the hub genes in the top 50 in intramodule connectivity (represented by dark-red ovals) and starch biosynthesis genes in the hub genes of the top 30% in intramodule connectivity (represented by clay-brown ovals) in the lightcyan (**b**) and brown (**c**) modules. Edge width indicates the weight of the relationship between two genes. (**d**,**e**) Pearson correlation between the genes in correspondence to (**b**,**e**). ** indicates that the correlation reached a significant level of 0.01. *FtPUL*, FtPinG0000055300.01; *FtNAC70*, FtPinG0002339300.01; *FtGBSS1-3*, FtPinG0000359400.01; *FtBE2*, FtPinG0000080700.01; *FtISA3*, FtPinG0009517500.01; *C3H*, FtPinG0003310400.01; *FtbHLH5*, FtPinG0000281400.01; *FtSS3-5*, FtPinG0003226800.01; *FtSS1*, FtPinG0005939600.01.

**Figure 6 ijms-23-15774-f006:**
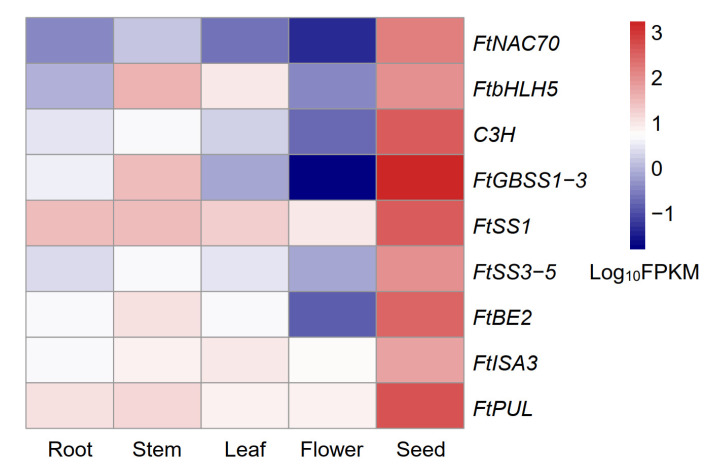
Tissue-specific expression patterns of nine candidate genes in the regulatory network of the starch biosynthesis pathway.

## Data Availability

The raw sequence data are deposited in the Genome Sequence Archive in the National Genomics Data Center, Chinese Academy of Sciences, under accession number CRA008415, which is publicly accessible at https://ngdc.cncb.ac.cn/gsa (accessed on 8 September 2022).

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
