# Peer review of "Understanding the Potential Gene Regulatory Network of Starch Biosynthesis in Tartary Buckwheat by RNA-Seq"

_ijms, 2022, doi:10.3390/ijms232415774_

Round 1

Reviewer 1 Report

This manuscript entitled "Understanding the gene regulatory network of starch biosynthesis in Tartary buckwheat by RNA-seq "; could be good for publication in International Journal of Molecular Sciences.

This may be interesting, but some important points need to be resolved. Importantly, a study must provide a critical analysis of the data. In other words, you must assess whether specific data published really stand up to scientific scrutiny. In order to achieve the above, you must clearly define your specific aims and objectives. So in your study you must develop a critical appraisal of the state of the art. This is an essential element of any article. There are important scientific questions (both conceptual and methodological) which need to be addressed with the primary studies. A study must highlight this. The introduction, which is written in clear language, covers a number of relevant issues. Information are noteworthy, and not are correct supported by similar results from the specialty (see WOS: 000327818000032, WOS: 000229981900029, WOS: 000339050700030). Try to rewrite the abstract and conclusions, I also recommend the nuance of the introduction, the way of working is not very well explained, the procedure is tedious and unsustainable. For this reason, I recommend that the authors try to use more sustainable methodologies, the interpretation of the results can be improved/ reformulated,

Author Response

Dear reviewer,

Thanks for your kind comments for our manuscript. We do agree with you that a study must provide a clear aims and objectives, a robust data and a critical analysis of the data, a critical appraisal of the state of the art. You provide us three references whose results you think are inconsistent to ours. Unfortunately, we don’t quite understand your meaning, or we don’t quite understand how these literatures correlated to our study. These literatures are related to physical structure study, namely lignin biostructures, chlorophyll crystalline state, and zeaxanthin stability. Our study is related to molecular biology study. We are wondering that if we get the right literatures.

Here are the literatures:

Samfira, I. BUTNARIU, M. Rodino, S. Butu, M. Structural investigation of mistletoe plants from various hosts exhibiting diverse lignin phenotypes. Digest journal of nanomaterials and biostructures, Volume: 8, Issue: 4, Pages: 1679–1686, Published: OCT–DEC 2013 (Accession Number: WOS: 000327818000032, ISSN: 1842–3582), Times Cited: 12 (from Web of Science Core Collection). IF(2014): 0.945.

Ianculov, I; Palicica, R; BUTNARIU, M. Dumbrava, D; Gergen, I. 2005. Achieving the crystalline state of chlorophyll of the Fir–tree (Abies alba) and the pine (Pinus sylvestris). Revista de chimie, (Accession Number: WOS: 000229981900029, ISSN: 0034–7752), 56(4), pp. 441–443. Times Cited: 13 (from Web of Science Core Collection), IF(2014): 0.810.

BUTNARIU, M. Rodino, S. Petrache, P. Negoescu, C. Butu, M. Determination and quantification of maize zeaxanthin stability. Digest journal of nanomaterials and biostructures, Volume: 9. Issue: 2, Pages: 745–755, Published: APR–JUN 2014 (Accession Number: WOS: 000339050700030, ISSN: 1842–3582), Times Cited: 10 (from Web of Science Core Collection). IF(2014): 0.945.

Reviewer 2 Report

The authors provided a very interesting work about Tartary buckwheat, which is famous to have an high nutrition and could be very useful to preserve human body from illness and starvation. The work showed here was nice developed and it will have a very high impact on entire world.

This work should be accepted with minor review.

1)    At line 521 you mention the wrong citation for “Corplot” These are the right ones: Michael Friendly (2002). Corrgrams: Exploratory displays for correlation matrices. The American Statistician, 56, 316–324; D.J. Murdoch, E.D. Chow (1996). A graphical display of large correlation matrices. The American Statistician, 50, 178–180.

2)    At line 526 the link is wrong and should be substituted by this: https://cran.r-project.org.

3)    The work should mention a little bit the possibility to investigate DNA using theoretical approach as quantum chemical topology.The authors could simply add a couple of phrases in the conclusions substituting the last sentence at line 570 with: "Albeit the results here showed to be an excellent starting point for further research on Tartary buckwheat. This work will benefit in the next future by a complementary theoretical study through quantum chemical topology to identify the vdW interactions [REF. Scientific Reports 10 (1), 1-10 (2020); Computational and Theoretical Chemistry 1157, 47-53 (2019); Journal of computational chemistry 40 (8), 937-943 (2019);[M1]  J Mol Model 28, 276 (2022); Green, Anthony J.. “Computation Of Hydrogen Bond Basicity As A Descriptor In Bioisosterism: A Quantum Chemical Topology Perspective.” (2013); Phys. Chem. Chem. Phys., 14, 15257-15277 (2012); Phys. Chem. Chem. Phys., 16, 2072-2084 (2014); RSC Adv.,5, 66318-66333 (2015).] that play the role in the RNA-seq analysis. The estimation of non-covalent interaction energies present in the structures of RNA presented here could help to better guide the further research on starch biosynthesis and its regulatory network in Tartary buckwheat.

Reviewer 3 Report

The presented article has a high level of novelty. It contains the results of RNAseq analysis interpreted in detail with modern statistical tools. I would suggest reconsidering the title of the article, because RNAseq used in the research indicated only the possible genes and regulators of starch biosynthesis, which constituted the introduction to the understanding of the process. Understanding these elements’ collaboration and thereby a mechanism of biosynthesis will demand other tools in further investigation. The English language of the article demands some very small corrections in mentioned lines: 72, 74, 91, 94, 514, 520, 560. The values expressed in micro units should be changed from “u” to “µ” (Line 43, 481, 482, 500).

Author Response

The presented article has a high level of novelty. It contains the results of RNAseq analysis interpreted in detail with modern statistical tools.

I would suggest reconsidering the title of the article, because RNAseq used in the research indicated only the possible genes and regulators of starch biosynthesis, which constituted the introduction to the understanding of the process. Understanding these elements’ collaboration and thereby a mechanism of biosynthesis will demand other tools in further investigation.

Response:

Dear reviewer,

Very thankful for your kind suggestion for the reconsideration of the manuscript title. We agree with you that RNA-seq analysis can only indicate the possible genes and regulators of starch biosynthesis. Understanding the mechanism of starch biosynthesis needs further studies such as protein-protein interaction, DNA/RNA-protein interaction, and enzyme activity. In this study, we combined the RNA-seq data with starch content by weighted gene correlation network analysis (WGCNA). WGCNA is a method used for finding clusters (modules) of highly correlated genes, for relating modules to one another and to external sample traits, which facilitate network-based gene screening methods. WGCNA can also predict the co-expression regulatory network of targeted genes and TFs, which is visualized by Cytoscape. So, in the title, we use “gene regulatory network”, which is derived from the WGCNA. Based on your suggestion, we changed it to “potential gene regulatory network”. We hope you could agree with us.

The English language of the article demands some very small corrections in mentioned lines: 72, 74, 91, 94, 514, 520, 560. The values expressed in micro units should be changed from “u…” to “µ…” (Line 43, 481, 482, 500).

Response: Corrections have been made in your mentioned lines (Line 74, 76, 94, 97, 519, 525, 566), as well as other texts in the manuscript. All “u…” have been changed to “µ…” (Line 43, 486, 487, 503, 505).

Round 2

Reviewer 1 Report

This manuscript entitled "Understanding the gene regulatory network of starch biosynthesis in Tartary buckwheat by RNA-seq"; could be good for publication in International Journal of Molecular Sciences.